# Colorectal Cancer and Purinergic Signalling: An Overview

**DOI:** 10.3390/cancers14194887

**Published:** 2022-10-06

**Authors:** Gabriela Gonçalves Roliano, Juliana Hofstätter Azambuja, Veronica Toniazzo Brunetto, Hannah Elizabeth Butterfield, Antonio Nochi Kalil, Elizandra Braganhol

**Affiliations:** 1Programa de Pós-Graduação em Ciências da Saúde, Universidade Federal de Ciências da Saúde de Porto Alegre (UFCSPA), Porto Alegre 90050-170, Brazil; 2Department of Pediatrics, University of Pittsburgh School of Medicine, Pittsburgh, PA 15213, USA; 3Programa de Pós-Graduação em Biociências, Universidade Federal de Ciências da Saúde de Porto Alegre (UFCSPA), Porto Alegre 90050-170, Brazil; 4Instituto de Cardiologia do Rio Grande do Sul-Fundação Universitária do Instituto de Cardiologia (IC-FUC), Porto Alegre 90040-371, Brazil; 5Serviço de Cirurgia Oncológica, Santa Casa de Misericórdia de Porto Alegre (ISCMPA), Porto Alegre 90020-090, Brazil

**Keywords:** colorectal cancer, purinergic signaling, immune system, tumor microenvironment

## Abstract

**Highlights:**

**What are the main findings?**
CD39-CD73 axis in tumor-associated immune cells promotes immune exhaustion, impairment of antitumor immune activity, and increased CRC progression.CD73 overexpression in cancer cells is associated with tumor growth, chemoresistance, and decreased patient survival.Extracellular ATP exhibits dual effects, either reducing tumor cell proliferation or favoring chemoresistance in a concentration-dependent fashion.Purinergic signaling components exhibit prognostic value and have the potential to be utilized as therapeutic targets.

**What is the implication of the main finding?**
CD73 induces colorectal cancer progression.CD39 contributes to immunosuppressive activity of T cells and poor prognosis.P2 and P1 purinoceptors were mainly associated with poor prognosis for patients.Ectonucleotidases and purinoceptors have potential as therapeutic targets in CRC.

**Simple Summary:**

Colorectal cancer is a leading cause of cancer-related death. Activated immune cells have the potential to eliminate tumor cells, but cancers gain the ability to suppress immune cell functions and escape immune attack. We explored one mechanism that cancers use to evade immune cells in colorectal cancer. This mechanism alters levels of molecules known as purines. Purines are key players in cellular energetics and many cellular processes and can also lead to immune suppression in cancer. In this study, we reviewed the literature that identified specific enzymes (ectonucleoetidases) and receptors (purinoceptors) which alter the levels of purines and contribute to immunosuppression in colorectal cancer. We hope that targeting these enzymes and receptors can activate the immune response against colorectal cancer and halt cancer growth and progression.

**Abstract:**

Colorectal cancer (CRC) is among the most common cancers and exhibits a high fatality rate. Gut inflammation is related to CRC, with loss of homeostasis in immune cell activities. The cells of the innate and adaptive immune system, including macrophages, neutrophils, mast cells, and lymphocytes, are present in most solid tumors. Purinergic signaling allows for communication between immune cells within the tumor microenvironment (TME) and can alter the TME to promote tumor progression. This system is regulated by the availability of extracellular purines to activate purinoceptors (P1 and P2) and is tightly controlled by ectonucleotidases (E-NPP, CD73/CD39, ADA) and kinases, which interact with and modify nucleotides and nucleosides availability. In this review, we compiled articles detailing the relationship of the purinergic system with CRC progression. We found that increased expression of CD73 leads to the suppression of effector immune cell functions and tumor progression in CRC. The P1 family purinoceptors A1, A2A, and A2B were positively associated with tumor progression, but A2B resulted in increased cancer cell apoptosis. The P2 family purinoceptors P2X5, P2X7, P2Y_2_, P2Y_6_, and P2Y_12_ were factors primarily associated with promoting CRC progression. In summary, CD39/CD73 axis and the purinergic receptors exhibit diagnostic and prognostic value and have potential as therapeutic targets in CRC.

## 1. Introduction

Colorectal cancer (CRC) is the third most common cause of cancer-related death worldwide [1]. CRC is characterized by two main mechanisms of gene instability: somatic mutations to the DNA that generate chromosomal instability (84% of CRC cases) and defective mismatch repair (MMR) processes that generate microsatellite instability (MSI) (13% of CRC cases) [2]. Among these two mechanisms, MSI is a predictive marker for CRC patients to have a positive response to chemotherapy and immunotherapy [3].

CRC classification is performed by analyzing tissue biopsy obtained from surgery following the TNM staging criteria, namely tumor size and invasion (T), spread to lymph nodes (N), and metastasis (M) [4]. Surgery without adjuvant therapy is typically indicated for stage I CRC while the more advanced stages (stage III and IV) require chemo/radiotherapy due to more advanced tumor progression [5,6]. Further, 5-fluorouracil, a DNA intercalating agent, is the first line chemotherapy agent for advanced CRC [7,8]. Monoclonal antibodies targeting VEGF and EGRF can also treat advanced CRC, but their use is provider-dependent [9,10].

Despite various efforts to develop effective therapies for CRC, distant metastasis is the most common cause of death for this patient population [11,12]. Patients with metastatic CRC have a relative five-year survival of 14% [13]. TNM staging can lead to the over- or undertreatment of CRC due to the high molecular heterogeneity of the cancer [14]. Consequently, Marks et al. and Puppa et al. propose including new cellular and molecular markers of CRC with histological characteristics to aid in cancer prognosis and patient follow-up during treatment [15,16].

The increased need for cellular markers of CRC highlights the importance of investigating immune cells involved in CRC progression [17]. Immune cells likely have a role in CRC development since chronic inflammatory diseases related to the gastrointestinal tract (GI), such as Crohn’s disease (CD) and ulcerative colitis (UC), increase the risk of developing CRC [18,19]. Immunosuppressive environments that favor cancer progression can be created through dynamic communication between non-transformed microenvironmental cells and cancer cells [20,21,22].

Purinergic signaling is an example of the aforementioned communication that can drive cancer progression since it integrates the immune/inflammatory responses with tumor proliferation/migration events [23,24,25]. The purinergic system is characterized by a signaling cascade mediated by purine-based nucleotides and nucleosides in the extracellular space [26,27]. These extracellular nucleotides and nucleosides, such as ATP and adenosine, regulate fundamental immune cell activities via purinoceptor sensitization [28,29]. Specifically immune functions purinergic signaling assists in include migration, polarization, clonal expansion, and cytokine production.

Considering the fact that immune cell dysfunction is related to CRC progression and metastasis, we hypothesized that alterations in purinergic signaling may contribute to cancer-associated immune dysfunction in CRC. Therefore, we reviewed the literature to identify studies that examine the role of immune cells in CRC-related inflammation. Additionally, purinergic signaling in CRC cells and tumor-associated immune cells is discussed to highlight purinergic signaling alterations with the most potential to provide diagnostic, prognostic, or therapeutic value. Greater understanding of the role of purinergic signaling in CRC progression and metastasis will create new avenues for future therapeutic interventions.

## 2. Methodology

Two independent reviewers searched the databases, Pubmed, Web of Science, ScienceDirect, and Scopus, to identify the publications that fit the inclusion criteria and a third reviewer resolved any discrepancies between the two reviewers.

The following search terms were used when searching the four databases: “colorectal cancer” in combination with each of the following terms “CD73;” “CD39;” “NTPDase1;” “ecto-5′-nucleotidase;” “purinergic signaling;” “adenosine;” “adenosine triphosphate;” “purinergic receptors;” “inflammation and purinergic signaling;” “metastasis and purinergic signaling;” “A1 receptor;” “A2A receptor;” “A2B receptor;” “A3 receptor;” “P2X receptor;” “P2Y receptor.”

Only full text articles published between January 2009 to September 2022 in English were included. Thirty-two of the publications were excluded because they were abstracts, book chapters, review articles, in other languages different from English or with access not publicly available. Finally, 59 articles were included in this review. A total of 1134 papers were initially identified among the four databases. A total of 491 publications remained after duplicates were removed. Among the 491 publications, 107 were selected following the evaluation of the title and abstract. In the end, 62 articles of the 107 selected fit the inclusion criteria and were utilized for the review. Once the final 62 publications were identified one of the authors extracted the pertinent variables from each of the publications (Figure 1). The variables extracted briefly included the material studied, sample size, target, and main results. A second reviewer reviewed the variables extracted by the initial review to ensure there were no errors or missing data. The data investigated provided information about purinergic receptors and enzymes, immune cells, and tumor cells.

## 3. Immune Cell Dysfunction Is Associated with CRC Progression

The cells of the innate and adaptive immune system are critical regulators of tumor growth. These cells include macrophages, neutrophils, mast cells, and lymphocytes and are present in most solid tumors, contributing substantially to tumor mass [30,31]. These cells mediate inflammatory responses that can either constrain or promote tumor progression [32].

During initial phases of tumorigenesis, NK cells and CD8+ lymphocytes recognize and eliminate immunogenic transformed cells. However, those cells that evade immune detection persist, selecting for continued growth and establishment of less immunogenic cancer cell variants [33]. As cancer progresses, immune cells are recruited to the tumor microenvironment (TME) through various soluble factors released by tumor cells, such as cytokines and chemokines. However, avoiding immune destruction is a hallmark of cancer, and tumors re-educate infiltrating immune cells towards anergy and immunosuppressive phenotypes that further promote tumor growth [32]. High levels of T cell infiltrate correlate with improved prognosis in multiple solid tumors, whereas macrophage levels often correlate with worse prognosis [33]. In sum, the nature of the immune infiltrate exerts considerable effects on tumor progression.

Intestinal epithelial cells (IECs) play a critical role in the delicate balance of intestinal homeostasis [34]. The lumen of the GI tract is not sterile, and therefore the bowel must exhibit an exquisite system that is both tolerant to commensal microbiota and other foreign contents, but simultaneously capable of recognizing pathogens [34,35]. IECs are vital to this system with roles both in promoting tolerance and in pathogen detection.

The GI microbiome contributes to host homeostasis through production of vitamins, education of immune cells, and competition with invading pathogenic bacteria [36]. Secretory goblet cells and Paneth cells are specialized IECs which contribute to protection of the GI microbiome from the host immune system through mechanisms including production of mucins and antimicrobial proteins (AMPs). Mucins and AMPs exclude bacteria from the epithelial surface and provide a barrier between commensal microbes and underlying immune cells. Tolerance is also promoted through IEC production of cytokines, such as transforming growth factor-β (TGFβ) and retinoic acid (RA), which promote tolerogenic macrophage and dendritic cell (DC) conditioning. Toleragenic DCs subsequently crosstalk with adaptive immune cell populations and promote the differentiation of naive CD4+ T cells into regulatory T cells (Tregs) and B cells into IgA-producing plasma cells. In addition to promoting tolerogenic immunity, IECs promote appropriate immune responses against pathogens through mechanisms, including expression of pattern recognition receptors (PRRs), such as RIG-I-like receptors (RLRs) and Toll-like receptors (TLRs). These receptors contribute to host defenses through the recognition of microbial motifs and pathogen virulence properties and are regulated to discriminate between commensal and pathogenic microbes for appropriate scaling of inflammatory responses [34].

### 3.1. Influence of the Gut Microbiome on Tumorigenesis

The potential for the gut microbiome to directly contribute to tumorigenesis has recently been recognized [37]. Individuals with familial adenomatous polyposis (FAP), a syndrome that leads to sporadic colorectal cancer, were shown to have enrichment of biofilms containing *Escherichia coli* and *Bacterioides fragilis* compared to healthy individuals. Oncotoxins produced by these bacteria were also enriched in FAP patient mucosa relative to the mucosa of healthy individuals [37]. Non-carcinogenic subtypes of these bacteria are present in healthy mucosa, and this demonstrates the direct pro-tumorigenic potential of the gut microbiome [37]. Conversely, microbiome-derived signals, such as induction of type I interferon (IFN-I) production, can program mononuclear phagocytes in the TME toward immunostimulatory monocytes and DCs. Interestingly, manipulating microbiota with a high-fiber diet can improve the antitumor immune response [38].

### 3.2. Macrophages and Neutrophils

In breast, skin, and pancreatic cancer, the presence of extensive immune infiltrate is correlated with poor prognosis [39,40,41]. Macrophages and neutrophils are innate immune cells that are critical components of this tumor-promoting immune infiltrate.

Neutrophils defend against pathogens by degranulation and release of neutrophil extracellular traps (NETs) that promote pathogen immobility and aid phagocytosis [42]. In cancer, NETs increase adhesion of tumor cells, concentrating groups of tumor cells and promoting the degradation of extracellular matrix to facilitate tumor invasion and spread [33,43]. Tumor-associated neutrophils (TANs) also promote angiogenesis through MMP9 and VEGF secretion [44]. Additional tumor-promoting activities of TANs include contributing to T cell immunosuppression, induction of a chronic cancer-promoting inflammatory environment, and favoring the establishment of a pre-metastatic niche, all leading to increased tumor growth [45,46].

Tumor-associated macrophages (TAMs) produce cytokines that promote angiogenesis, T cell inactivation, Treg induction, inhibition of DC activity, chemoresistance, and increased cancer cell proliferation [47,48]. TAMs can also promote initiation of metastasis through tissue remodeling, promotion of epithelial-mesenchymal transition, and promoting increased deposition of NETs by TANs [33]. In sum, TAMs and TANs tend to favor immune escape and survival of tumor cells. However, TAMs and TANs can also exert anti-tumor effects, and are associated with better response to chemotherapy in CRC patients [49,50].

The phenotypic state of macrophages and neutrophils helps define the relative extent of their pro-tumor versus anti-tumor activity. Depending on the environment, macrophages and neutrophils can undergo classical (M1/N1) or alternative polarization (M2/N2). Whereas M1/N1 TAMs/TANs promote elimination of pathogens and immunogenic transformed cells, M2/N2 TAMs/TANs reduce inflammation and promote tissue homeostasis and repair [51,52]. Tumor cells often gain the capacity to “corrupt” TAMs/TANs to a M2-like/N2-like phenotype, which contributes to tumor progression [53,54,55].

### 3.3. Myeloid-Derived Suppressor Cells

Myeloid-derived suppressor cells (MDSCs) are a group of heterogeneous cells that originate in the bone marrow. These were recently described as potent immunosuppressive agents in cancer and other conditions, such as chronic inflammation and infections [56,57].

### 3.4. Lymphocytes

NK cells are innate lymphocytes which perform immune surveillance and operate mainly by recognition of major histocompatibility class I (MHC-I) expression [58,59]. MHC-I expression is frequently downregulated in tumor cells, allowing these cells to evade detection by NK cells. Additional mechanisms that tumor cells utilize to evade NK cell responses include (1) cleavage of stimulatory ligands and (2) secretion of immunoregulatory molecules such as adenosine (ADO), TGF-α and prostaglandin E2 [60,61,62,63,64]. Low NK cell levels and activity have been reported in various types of solid tumors, including CRC [65,66,67].

T cells are major players in the adaptive immune response and are divided into subpopulations [68]. Effector T cells include CD8+ T cells, which exert cytotoxic activity against target cells, and CD4+ T helper cells, which produce inflammatory cytokines that amplify antitumor responses through macrophage and NK cell activation [69,70]. Increased effector T cell infiltrate can be associated with better prognosis for cancer patients, including those with CRC [69,71]. However, similar to their immunosuppressive actions towards TAMs and TANs, tumor cells often reduce activity of effector T cells and promote the recruitment of immunosuppressive CD4+ Tregs [68]. Tregs (CD4+CD25+FOXP3+) are responsible for immunological self-tolerance and for tissue homeostasis during inflammation, decreasing the number and activity of effector immune cells and stimulating tissue repair [72]. Tregs suppress anti-tumor effector cells, including CD8+ T cells, CD4+ T cells, macrophages, NK cells, and neutrophils, through a variety of mechanisms [72,73]. These mechanisms include contact with other cells through inhibitory receptors CTLA4, PD-1, PDL-1 and LAG3 [74,75], as well as production of immunosuppressive molecules, such as IL-10, TGFα, and ADO [76]. Treg infiltration has been associated with tumor progression and worse prognosis in various types of cancer [77]. However, the association between Treg infiltration and prognosis in CRC is not yet understood, and studies have yielded conflicting results [50,78,79,80,81].

Exhausted CD8+ T cells (Tex) are another T cell subpopulation that has been extensively studied in inflammation, infection and cancer. After long-term stimulation, effector T cells lose their ability to proliferate and produce inflammatory cytokines [82,83,84]. Tex cells are characterized by the high and persistent expression of inhibitory receptors such as PD-1, which represents a therapeutic opportunity, as function can be restored through blockade of PD-1 interaction with its ligand PDL1 [85,86]. Tex are abundant in the TME of CRC and these have potential use as a biomarker and therapeutic target [87,88,89].

Immune checkpoint molecules, such as PD1 and CTLA4, are targetable components of T cells in the TME. The blockade of PD-1/PDL-1 and CTLA4 using monoclonal antibodies was the first strategy employed for cancer-based immunotherapy and has benefitted many patients [90,91,92,93]. However, a variety of tumors types with different mutations profiles are not sensitive to checkpoint inhibitor therapy, and these types of therapies have proven challenging in CRC [94,95,96].

In summary, tumor cells induce escape elimination by the immune system through a variety of mechanisms. These include evasion of NK cell detection through decreased MHC-I antigen presentation, promotion of an immunosuppressive TME by recruiting Tregs, polarization of TAMs/TANs towards immunosuppressive phenotypes, secretion of anti-inflammatory cytokines (IL-10, TGFα), and expression of ligands for inhibitory receptors (PD-1, CTLA4). It is critical to understand the complex interplay between tumor cells and the TME in order to effectively re-educate immune cells towards anti-tumor functions. In addition to the mechanisms described above, the purinergic system is a key player that controls tumor-implicated immune responses and cancer progression. These purinergic mechanisms will be the focus of the remainder of this review.

## 4. Purinergic Signaling in Colorectal Cancer

### 4.1. Ectonucleotidases—General Aspects

Purinergic signaling involves the biological effects of extracellular purines and pyrimidines on purinoceptors. This mechanism is tightly controlled by ectonucleotidases, enzymes that aid in the conversion of adenosine triphosphate (ATP) to ADO within the extracellular space [26]. The role of ectonucleotidases in the production of immunosuppressive ADO maybe critical to understanding the progression of CRC. ADO can be produced via two adenosinergic pathways: the canonical and non-canonical pathways. The canonical adenosinergic pathway includes enzymes that convert ATP to AMP (Figure 2). These enzymes include ecto-nucleoside triphosphate-diphosphohydrolases (E-NTPDases), such as the NTPDase1, also known as CD39, which converts ATP to ADP and ADP to AMP, ecto-pyrophosphate-phosphodiesterases (E-NPP) which convert ATP to AMP, alkaline phosphatases (ALPs) which convert ATP to ADP, ADP to AMP, and AMP to ADO, ecto-5′-nucleotidase/CD73 (CD73) which converts AMP to ADO, and adenosine deaminase (ADA) which converts ADO to inosine [27,97,98]. ADO can also be generated by the non-canonical adenosinergic pathway, which involves the enzymes nicotinamide adenine dinucleotide (NAD+)-glycohydrolase/CD38 which converts NAD+ to ADP-ribose (ADPR), and NPP1/CD203a (PC-1) which converts ADPR to AMP. This final step generates extracellular AMP which is metabolized to ADO by CD73. Therefore, CD73 represents the common link between the canonical and non-canonical adenosinergic pathways [99].

Extracellular ADO levels are also regulated by nucleoside equilibrative transporters (ENTs) and concentrative nucleoside transporters (CNTs). ENTs and CNTs are expressed in the cell membrane and help transport ADO into the cells [24,100]. Within the cell, ADO is phosphorylated by ADO kinase (AdoK) and adenylate kinases for subsequent conversion into ADP [25]. ADO can also be deaminated to inosine (INO) by adenosine deaminase (ADA) [97,98]. ADA is expressed in most human tissues, with the highest levels found in the lymphoid system, including lymph nodes, spleen, and thymus [101]. ADA binds to the cell surface through an ADA binding protein termed CD26 [102]. The complexity of extracellular ATP metabolism by ectoenzymes has been increasingly appreciated and reviewed in last years [27]. In this review, the purinergic signaling participation in CRC progression was addressed. The main findings were summarized in the Table 1.

#### 4.1.1. Ectonucleotidases—The Role of CD39 in CRC Progression

Several studies have investigated the role of Tregs in the TME of CRC with the aim of characterizing Treg subpopulations and their immunosuppressive potency [103,104,105]. Multiple lines of evidence highlight CD39+FOXP3^+^ T cells as the as the major regulatory T cell infiltrate in tumor tissue. Furthermore, Tregs are also elevated in peripheral blood from patients with CRC compared to healthy donors [35,103,104,105,106,107,108]. Levels of CD39+Tregs in peripheral blood from CRC patients correlate with TNM staging and clinicopathological features, and the expression of CD39 gradually increases in Tregs from initial to advanced stages of CRC development [104,106,109]. Similar to FOXP3, Helios is a marker that characterizes immunosuppressive Tregs [110] and its co-expression with CD39 was found in multiple studies [105,108]. Additionally, CTLA-4 and PD-1 checkpoint inhibitor molecules were co-expressed with CD39 in Tregs from both CRC tumor bulk and the peripheral circulation, suggesting that this lymphocyte subpopulation participates in CRC-associated immunosuppression [103,104,105]. In line with this, Khaja et al. proposed simultaneous blockade of CD39 and PD-1 as a new modality of treatment for patients with CRC (2017).

In addition to demonstrating the presence of Tregs in CRC samples, the immunosuppressive activity of CD39+ Tregs was also investigated. CD39+ γδTregs (a subtype of T cells located mainly in the GI tract) isolated from CRC biopsies promoted a higher levels of inhibition of in vitro CD3^+^ effector T cell proliferation and function when compared to other T cells subsets extracted from the same tissue [104]. ADO receptor blockade dramatically reversed this immunosuppressive effect, while inhibition of IL-10, TGF-β, CTLA-4, and PD-1 signaling did not substantially change levels of Treg-induced immunosuppression [104]. Adenosinergic signaling has also been reported to mediate differentiation of tumor-infiltrating CD39+ γδTregs induced by tumor-derived TGF-β1 [104].

The primary location of CRC has also received considerable attention, due to differences in cellular/molecular characteristics, response to therapy, and patient outcomes based on location in the right (ascending) or left (descending) colon [111]. Several lines of evidence point towards worse outcomes in right-sided compared to left-sided CRC, causing location to be classified as a patient risk factor [112]. In line with the role of CD39 in tumor promotion, one group observed an increased frequency of CD39+ γδTregs in right-sided compared to left-sided CRC [113]. Additionally, CD39 expression in γδ-TILs increased in vivo tumor growth, metastasis, and invasion [113]. In human clinical biopsies, low CD39 expression in right-sided CRC was correlated with improved patient outcomes, while in left-sided CRC low CD39 expression was associated with worse patient outcomes [113]. High frequency of CD39 positivity in Tregs positively correlates with Treg immunosuppressive activity, which can be reversed by siRNA knockdown of CD39 [106].

CD39^high^ Tregs can also suppress T cell migration in CRC. One study found that the CD39^high^ Treg subpopulation derived from CRC patients reduced CD4+ and CD8+ T cell transendothelial migration (TEM), whereas CD39^high^ Tregs derived from healthy volunteers did not exert these effects on TEM. CD39 enzyme activity contributes to ADO-mediated reduction of T cell migration, as addition of ADO reduced TEM, whereas blockade of CD39 or ADO receptors enhanced TEM. Furthermore, the effects of ADO on TEM were shown to be mediated by monocytes, as the addition of ADO decreased the ability of monocytes to activate the endothelium. Cellular expression of CD73 was low and did not differ between patients and controls [103]. Although more studies of the effects of CD39 expression on Treg CD39^high^ cells in CRC are needed, these data indicate that this is an important regulator of immunosuppression in the TME, with the ability to reduce T effector cell migration into the tumor.

CD39 expression was also studied in non-regulatory T cell subsets derived from CRC patients. Two different subsets of CD8+ T cells were identified in CRC tumors, one referred to as the bystander population, which did not exhibit tumor-specific antigen specificity, and the other referred to as tumor-specific TILs, which exert a tumor antigen-specific response. A marked lack of CD39 expression was demonstrated in the CD8+ bystander population (CD8+CD39^−^), whereas tumor-specific CD8+ TILs expressed high levels of CD39. The CD8+CD39+ TILs exhibited high expression of genes related to proliferation and exhaustion, which is characteristic of chronically stimulated T cells and indicates that these are an exhausted T cell population. Furthermore, CD8+CD39+ TILs expressed genes related to antigen presentation and processing. The authors proposed the use of CD39 as a biomarker for CD8+ T cells, where the lack of CD39 expression could be used to identify T cells with bystander roles [114]. Increased CD8+CD39^high^ TILS were also found in tumor compared to surrounding non-tumor tissue at initial stages of CRC (I-II). These cells were characterized by high PD-1 expression and low INFγ production, which was correlated with exhausted T cell phenotypes and suppressed CD4+ T cell proliferation [115]. In tumor samples, TIL CD4/CD8 presented a CD39 increase and CD73 decrease, suggesting that CD39 expression can be an inhibitor of T cells. CD39 was similar in mucosa, tumor, and tumor margin, suggesting that the other cells can express this enzyme. In contrast, CD39 was decreased and CD73 increased in advanced TNM, suggesting that advanced tumors had CD39 missed and CD73 overexpressed. The CD39 inhibition in autologous spheroids of CRC increased T cell capacity to attack these cells, by increasing movement and the destruction of spheroids [116].

Recently, a specific mucosal-associated invariant T cell (MAIT cell) was investigated in CRC [117]. MAIT cells are capable of recognizing microbial metabolites and can exert effects on anti-tumor immunity. Exhausted MAIT cells demonstrated reduced polyfunctionality with regard to the production of important anti-tumor effector molecules, including IFN-γ and GrB. PD-1 blockade partially improved in vitro activation of tumor-infiltrating MAIT cells [117]. However, whether CD39 participates as a source of immunosuppressive ADO was not investigated in this context.

T cells have been the predominant immune cell type investigated in the context of cancer-associated purinergic signaling, primarily due to CD39/CD73 expression, which has been proposed as a marker for immunosuppressive Tregs. However, as described above, other immune cell populations are present in the TME and yield important contributions to the creation and maintenance of an immunosuppressive microenvironment [56]. One study sought to define the role of the CD39/CD73 axis in MDSC populations in CRC. MDSCs were investigated during the course of treatment of patients with advanced stage metastatic CRC (mCRC) receiving FOLFOX plus bevacizumab therapy. A higher level of circulating MDSCs, in particular granulocytic MDSCs (gMDSCs), was observed in untreated mCRC and were associated with decreased overall survival (OS) and progression-free survival (PFS). FOLFOX plus bevacizumab treatment reduced gMDSC levels and increased OS and PFS. This study also investigated the expression of PD-L1, CD39, and CD73 in MDSCs, and found that gMDSCs from mCRC patients that expressed these markers exhibited potent immunosuppressive effects relative to other myeloid cell populations present in the blood. These immunosuppressive effects could be reversed by a blockade of the CD39/CD73 and PD1/PD-L1 axes, suggesting the potent immunosuppressive activity of these cells in CRC in an ADO-pathway and PD1/PD-L1 dependent process [118]. In accordance with other studies, the authors suggest the use of ATP ectonucleotidase inhibitors and/or PD-1/PDL1 inhibitors as potential therapeutic combinations with FOLFOX-bevacizumab [118]. An additional study sought to characterize MDCSs in CRC and demonstrated that the number of Lin^−/low^HLA-DR^−^CD11b^+^CD33^+^ MDSCs in peripheral blood were markedly increased in CRC patients compared to healthy donors. MDSCs exhibited elevated CD39 expression and were positively correlated with tumor metastasis. MDSCs cells also demonstrated potent immunosuppressive activity, including the ability to inhibit CD3^+^ T cell proliferation from two stage IV CRC patients [119].

The role of CD39 in CRC progression and metastasis was investigated in CD39^+^^/−^ mice, CD39^+^^/+^ mice, or over-expressing CD39 transgenic mice (htCD39) all in a BALB/c background. No differences in primary tumor growth were observed between these groups. However, in a metastatic dissemination model, tumors derived from the MC-26 murine colon adenocarcinoma cell line grew significantly faster in htCD39 mice compared to CD39^+^^/−^ mice, suggesting that CD39 overexpression in the TME favors tumor spread. In both primary and metastatic CRC models, CD39 was expressed at high levels around the tumor borders, including expression in stromal cells, endothelial cells, and immune cells [120]. In a separate study, increased CD39 expression was also reported in tumor tissue from CRC patients when compared to normal border, and in vitro inhibition of CD39 impaired tumor cell proliferation [121]. In addition, the presence of variant allele CD39 in patients with mCRC was an indicator of favorable response to chemotherapy plus bevacizumab, suggesting that this may be used as a predictive marker [122].

In a conflicting study, clinical CRC biopsies showed lower CD39 levels when compared to non-neoplastic tissue, particularly in early stages of tumor development. However, throughout tumor progression, CD39 levels became more comparable to levels on normal border (T3N ± M1). Additionally, P2Y_2_ and P2X7 expression was markedly decreased in tumor tissues when compared to normal borders [120].

In sum, multiple lines of evidence point to a strong association between CD39 expression and a suppressive immune cell profile, which was mainly characterized in Tregs but also extended to MDSCs. Nevertheless, the mechanisms underlying CD39-mediated immunosuppression, and its correlation with tumor progression and prognosis in CRC, remain to be more deeply understood. Further cell-type specific studies are needed to more fully characterize how CD39 expression in tumor, stromal, and epithelial cells within the tumor and the tumor borders impacts CRC progression and metastasis [120].

#### 4.1.2. Ectonucleotidases—Participation of CD73 in CRC Progression

CD73 is an enzyme that can be anchored to the plasma membrane in the C-terminal portion by a residue of glycosyl-phosphatidylinositol (GPI), or can be found in soluble form in the cytosol or the extracellular medium [100,123]. Physiologically, CD73 is widely distributed in different tissues, including the colon, kidney, brain, liver, heart, lung, vascular endothelium, spleen, lymph nodes, and bone marrow [124]. In the immune system, CD73 is found on the surface of Tregs, neutrophils, MDSCs, DCs, NK cells, and macrophages [25]. An increasing number of studies point to a role for CD73 in cancer pathogenesis.

CD73 expression was reduced in a highly liver-metastatic human CRC cell subline (SW48LM2 cells) when compared with less malignant tumor cells, both in normoxic and hypoxic conditions. In addition, metastatic CRC cells (mCRC) exhibited a decrease in extracellular nucleoside levels (ADO, guanosine and INO) and a parallel increase in nucleotides (AMP, GMP and IMP), which indicate that loss of CD73 expression and subsequent changes in purine metabolism are beneficial for tumor progression [125]. In contrast, low expression of CD73 was associated with improved progression free survival (PFS) in patients that received single or combined therapy with cetuximab or FOLFIRI/FOLFOX [126,127]. KRAS status also impacts the predictive role of CD73 in CRC therapy efficacy. For example, CD73 levels were predictive of PFS and OS benefit for cetuximab therapy of KRAS wild type tumors, while no predictive effects for CD73 were observed in KRAS mutant tumors [127].

In contrast, higher expression of CD73 in CRC has been reported in several studies [128,129,130,131,132]. Pathologically, CD73 is overexpressed in several types of tumors, including CRC [133,134,135,136,137,138]. CD73 overexpression was associated with increased cell proliferation [130], nerve invasion, lymph node and distant metastasis, and advance tumor staging [131]. In liver metastasis, CD73 expression was also associated with worse pathological features and poorer response to preoperative therapy [132]. CD73 expression was significantly lower in chemotherapy-responsive CRC patients, and RAS-MAPK-inhibition induces CD73 upregulation and an immunosuppressive TME [139]. CRC patients with CD73^high^ tumor expression have shorter OS when compared to those with CD73^low^ expression [128,129,131,132,139]. For these reasons, some authors suggest CD73 as a biomarker for poor prognosis [129,130].

CD73 has also been associated with CRC development and progression. CD73 inhibition in a mouse model of colitis-associated tumorigenesis reduced histologic evidence of colon damage. Furthermore, using the same model, the CD73 inhibitor AB680 unlocked the anticancer immune response, leading to increased CD8+T cell activation and improved Treg and exhausted T cell function (Kim et al., 2021). In a study with a microRNA termed miR30a that inhibits in vitro and in vivo CRC cell proliferation and promotes apoptosis, the authors suggest that the antitumor effects of miR30a are mediated by CD73 downregulation [140]. Furthermore, cancer associated fibroblasts (CAFs) express high levels of CD73 in CRC and are strongly correlated with poor prognosis [22]. In this paper, the authors demonstrated that elevated ADO upregulates CD73 via an A2B-mediated pathway, inducing an immunosuppressive TME [22]. Targeting this circuit significantly improved antitumor immunity in CAF-rich tissues [22]. The authors described CD73 as an immune checkpoint protein in a feedforward circuit, which is enhanced by tumor-derived ADO [22].

The levels of CD73 in CRC have also been correlated with macrophage phenotypes. A study showed that alternate-day fasting for two weeks led to a subsequent decrease in in vivo CRC growth and decreased M2-TAM polarization. In vitro, fasting conditions of low-glucose/low serum-containing media induced autophagy, CD73 suppression, and decreased generation of extracellular ADO in the CT26 mouse CRC cell line. Decreased ADO levels led to impairment of M2-TAM polarization and was associated with inactivation of JAK1/STAT3 pathway under fasting conditions, suggesting that antitumor immunity induced by fasting is mediated by blockade of ADO signaling [141].

Overall, these studies indicate that CD73 expression promotes CRC growth and metastasis. However, the tumor-promoting effects of CD73 are dependent on tumor stage and on its cell-type specific expression levels within the TME. Finally, considering that the majority of the above investigations were focused on lymphocytes, additional studies to the CD73 expression and function in innate immune cells present in the TME of CRC are also needed.

### 4.2. Purinoceptors—General Aspects

Purinergic signaling is initiated by the release of nucleotides and nucleosides into the extracellular space. Under normal conditions, ATP is in the millimolar range in the cytoplasm (3–10 mM) and in the nanomolar range in the extracellular space [142]. However, ATP can be released through the plasma membrane by lytic (cell death) and nonlytic mechanisms (vesicles and membrane channels formed by the pannexin-1/connexins) [143,144]. Following release into the extracellular space, purinergic nucleotides and nucleosides exert their effects through interactions with specific membrane receptors called purinergic receptors or purinoceptors [145]. These receptors are responsible for purinergic message transmission. Purinoceptors are divided into two groups: P1 receptors (P1Rs) and P2 receptors (P2Rs). P1Rs utilize the main endogenous agonist ADO, whereas P2Rs are sensitive to multiple di- and triphosphate nucleosides, such as ATP, ADP, UTP, and UDP [146].

#### 4.2.1. ATP as an Agonist of Purinoceptor-Mediated Protumor and Antitumor Actions

The role of extracellular ATP in CRC is still not well-understood, and the literature on this subject is conflicted. This may be due to different functions depending on the receptor and cell type involved. Multiple studies have shown that high concentrations of ATP and P2R agonists reduce cell viability and proliferation at the S phase of cell cycle through inhibition of protein kinase C (PKC) in CRC cell lines [147,148]. In contrast, a separate study indicates that exposure of Caco-2, a human CRC cell line, to ATP increases multidrug resistance-associated protein 2 (MRP2) expression, which was shown to confer resistance to etoposide, cisplatin and doxorubicin, leading to enhanced cell survival. It is possible that this mechanism is mediated by P2Y receptors, but further investigation is needed to elucidate which receptors may be involved [149].

The ATP-based chemotherapy response assay (ATP-CRA) is a method used to measure tumor response to therapy., since tumor cells quickly deplete ATP. In order to evaluate the utility of this assay in CRC, ATP levels were tested in peripheral blood samples from advanced CRC patients undergoing adjuvant chemotherapy with FOLFOX or Mayo clinic regimen (5-FU and leucovorin). Response for both treatments as indicated by ATP-CRA (>40% ATP reduction) positively correlated to OS and PFS, suggesting that the ATP-CRA is a useful test to guide individualized chemotherapy [150]. In contrast, in vivo tests with an immunogenic cell death (ICD)-inducing therapy resulted in enhanced ATP secretion, in addition to other danger signaling molecules, which correlated to tumor reduction and increased CD8+ T-mediated antitumor response [151]. Therefore, extracellular ATP levels may not be a consistent indicator of treatment responses.

The variety of pathways involved in ATP release to the extracellular environment, including cell membrane channels, cell membrane damage, or cell death, may explain the conflicting results in the above studies. To summarize, in vitro analyses point to the dual role of ATP, which can reduce CRC cell viability and proliferation [147,148] while simultaneously promoting chemoresistance [151]. Decreased ATP levels were a clinical indicator of chemotherapy responsiveness in CRC [150], whereas increased ATP secretion was reported in immunogenic cell death in vivo studies and was associated with an improved immunologic response [151]. Taken together, these studies leave open questions surrounding whether extracellular ATP benefits or harms CRC progression.

#### 4.2.2. P1 Receptors and Their Relationship with CRC Progression

P1Rs differ in their affinity for ADO. A1R, A2AR and A3R are high affinity receptors, whereas A2BR is a low affinity ADO receptor [152]. Under physiological concentrations, ADO is present at low levels in both the intracellular and extracellular space (nanomolar range) and mediates effects via A1R, A2AR, and A3R. When ADO concentrations reach elevated levels (micromolar range), such as under inflammatory conditions or in the TME, A2BR becomes the principal receptor that mediates ADO signaling [153]. Studies have shown that P1Rs are differentially expressed in CRC cell lines and tissue [154,155].

A1R and A3R are coupled to Gi or Go proteins and lead to decreased intracellular cyclic AMP (cAMP) levels, whereas A2AR and A2BR are coupled to Gs protein and result in increased levels of intracellular cAMP. Stimulation of A1R and A3R can trigger the release of Ca^2+^ ions from intracellular stores, whereas A2BR receptor stimulation can activate phospholipase C (PLC) [146,153]. All P1Rs are coupled to MAPK/ERK signaling pathways [156].

The A1R was investigated in one study involving the anti-diabetic drug metformin. Increased A1R expression was observed in HCT116 and SW480 CRC cell lines following treatment with metformin and corresponded with induction of apoptosis. This effect was dependent on AMPK-mTOR pathway, a key player in cancer cell survival/proliferation, which resulted in cell cycle arrest and apoptosis [154].

The A2BR exhibits a dual role in CRC progression. A2BR expression was considerably higher in CRC tissues compared to normal colonic mucosa and was present in five CRC cell lines (DLD1, SW480, HCT-15, LOVO, and COLO205). A similar cancer-specific increase was not detected for the other P1Rs (A1R, A2AR and A3R). Furthermore, A2BR was increased under hypoxic compared to normoxic conditions, whereas no increase in A1R, A2AR or A3R expression was demonstrated under hypoxic conditions [157]. It is possible that the overexpression of A2BR during hypoxia contributes to cancer cell growth and angiogenesis [157]. In contrast, the expression of A2BR was also associated with cell death in Saos-2 cancer cell line [158]. This study showed that activation of p53 in TetOn-p53-WT Saos-2 cells directly stimulates *A2BR* gene expression, which in turn contributes to p53-induced apoptosis. Moreover, A2BR was shown to be involved in hypoxia and chemotherapy-induced cell death in Saos-2 cells by regulating Bcl-2 family members, a group that mediates intrinsic cell death [158].

High A2AR expression has been demonstrated in CRC and associated with worse prognosis and the presence of tumor infiltrating lymphocytes (TILs) [155,159]. One study analyzing 204 tumor specimens from CRC patients demonstrated that both PD-L1 and A2AR expression was higher in tumor than in adjacent non-tumor tissue. Additionally, both PD-L1 and A2AR expression levels were correlated with higher TNM stage and lower OS, showing the independent prognostic predictor value of both immunosuppressive markers in CRC [155]. RNA sequencing and whole exome sequencing data from 453 CRCs demonstrated a positive association between TIL levels and expression of immune checkpoint molecules and A2AR in colon adenocarcinomas. These data are in accordance with the protumor role of the A2AR receptor [159].

High expression of A3R was demonstrated in HT-29 CRC cell line and in human CRC tissue compared to normal adjacent mucosa using [18F]FE@SUPPY as a PET-tracer for A3R [160]. Conversely, the A3R agonist (N6-(2,2-diphenylethyl)-2-hexynyladenosine) inhibited Caco-2 CRC cell proliferation in a concentration and time dependent manner. However, A3R knockdown did not reverse or prevent the antitumor effects induced by the A3 agonist, indicating that these effects were likely off-target [161]. Therefore, the role of A3R in CRC remains relatively unexplored.

#### 4.2.3. P2 Receptors—P2X Participation in CRC Development

P2Rs consist of two major categories of receptor, P2X1-7 ionotropic receptors that are sensitive to extracellular ATP and P2Y_1,2,4,6,11–14_ metabotropic receptors that are stimulated by ATP, ADP, UTP, UDP, and UDP-glucose [146]. P2XR receptors have intracellular N- and C-subunits that bind to protein kinases and also have two transmembrane regions, which are involved in channel activation. P2XR subtypes differ in their rates of desensitization, ion conductivity, and sensitivity to agonists, antagonists, and allosteric modulators [145,162].

The P2X1 receptor is widely expressed in smooth muscle cells; it also mediates the actions of ATP and Ca^2+^ influx on platelet aggregation [163,164]. P2X2 receptors are widely expressed in central and peripheral neurons and have been implicated in neurotransmission [165,166,167]. P2X3 receptors are expressed in sensory neurons [168,169,170]. The receptors P2X4 and P2X6 are expressed in the central nervous system, endothelial cells, and thymus [143,171,172,173]. The P2X7 receptor is expressed in immune cells, pancreas, skin, and microglia and leads to cell death in the presence of high levels of ATP [162,174,175,176,177,178,179,180]. P2XR cell signaling is mediated by gating of primarily Na^+,^ K^+^, and Ca^2+^ and, occasionally Cl^−^ channels [181,182].

P2X7 has been suggested as a target for CRC treatment [183,184]. Analysis of P2X7 levels in human tumor and adjacent tissue revealed that its expression was higher in tumors, and was an independent variable associated with worse prognosis, shorter survival, and higher TNM stage [184]. Increased P2X7 expression was found in mCRC tissues when compared to primary tumors, suggesting its association with metastasis. In line with this, higher P2X7 levels were also reported in CRC cell lines compared with normal colon cell lines, and in metastatic-derived CRC cell lines compared with primary tumor-derived cells [184]. In addition, induction of P2X7 signaling with use of a selective agonist resulted in PI3K/Akt and NF-κB pathway activation and further induction of CRC cell proliferation [183,184]. P2X7 overexpression also mediated in vivo tumor growth, by stimulating angiogenesis, cancer stem cell (CSC) properties, and macrophage/TAM infiltration [185]. Also consistent with promotion of CRC malignancy by P2X7, antagonism or knockdown of P2X7 impaired in vitro and in vivo cell proliferation, invasion, and migration and promoted apoptosis [186]. Interestingly, a recent study performed in a model of colitis-associated colorectal cancer revealed that P2X7 sensitization in combination with signals from a dysbiotic microbiota promotes CRC development by inducing the inflammasome activation and further inflammatory cascade amplification [187]. In contrast, P2X7 blockade in an in vivo colitis-associated cancer model promoted Treg and neutrophil infiltration and epithelial cell growth and decreased apoptosis, suggesting a protective factor of P2X7 against cancer formation [188].

Analysis of colon cancer tissue and normal colon tissue from two independent datasets (GEO and TCGA) indicated that P2X5 was among the top four high risk genes. Most of the differentially expressed genes that were identified are involved in protein transport, apoptosis and neurotrophin signaling pathways [189].

Out of all the P2X receptors listed, only papers on P2X7 and P2X5 were available for this review, demonstrating the lack of information about P2XR role in CRC biology. Overall, the available literature points to a role for these receptors in promoting CRC.

#### 4.2.4. P2 receptors—P2Y Participation in CRC Development

P2Y_1_, P2Y_12_, and P2Y_13_ are activated by ATP and ADP, P2Y_2_ and P2Y_4_ are activated by UTP, P2Y_6_ is activated by UDP, and P2Y_14_ receptor is activated by UDP-glucose [166]. P2YR-mediated activation of protein Gq leads to the stimulation of PLC, with subsequent production of inositol-(1,4,5)-trisphosphate and diacylglycerol (DAG). Inositol-(1,4,5)-trisphosphate leads to increased intracellular Ca^2+^ levels, and DAG stimulates PKC and may inhibit adenylate cyclase [190]. P2YRs are expressed in a variety of tissues and organs, including lung, kidney, pancreas, adrenal gland, heart, vascular endothelium, skin, muscles, and brain.

The purinergic receptors are very complicated to investigate, due to factors including methodologic limitations, complexity of receptor subtypes and activation, and cell-type specific mechanisms of action. Few studies of the P2YR were identified for this review and only included P2Y_2_, P2Y_6_, and P2Y_12_. The P2Y_2_ receptor was investigated in one in vitro study, which analyzed resistance to apoptosis mediated by treatment with ursolic acid in CRC HT-29 cells. Results demonstrated that ursolic acid treatment induced ATP production, which is probably released and binds to P2Y_2_ receptors. The P2Y_2_ receptor then activates tyrosine kinase Src, leading to p38 phosphorylation. The p38 pathway induces the expression of COX-2 and results in chemoresistance in HT-29 CRC cell line [191].

The P2Y_6_ receptor was analyzed in two articles, one in vitro and one both in vitro and in vivo. The role of P2Y_6_ receptors in cell migration was investigated in lung and colon cancer cell lines. Stimulation of a Caco-2 cell monolayer with a P2Y_6_-selective agonist (MRS2693) induced filling in the plaque of the cells and formation of focal adhesions and filopodia. Treatment with P2Y_6_ antagonist (MRS2578) had the opposite effect, demonstrating a potential role for this receptor in the tumor growth [192]. Meanwhile, treatment of HT-29 cells with the P2Y_6_ agonist prevented the pro-apoptotic effects of TNF-α, leading to increased expression of X-linked inhibitor of apoptosis protein (XIAP), which was correlated with Akt/PI3K phosphorylation [193].

In vivo experiments with P2Y_6_^+/+^ and P2Y_6_^−/−^ mice demonstrated that P2Y_6_^−/−^ animals had significantly reduced number and size of colorectal tumors. Besides this, P2Y_6_^−/−^ animals had a significantly lower dysplastic score and reduced vascularization when compared to P2Y_6_^+/+^ mice, indicating a potential role for this receptor in promoting tumorigenesis. Expression of β-catenin was investigated to further understand the influence of P2Y_6_ loss. β-catenin was observed in the plasma membrane of cells derived from P2Y_6_^−/−^ animals, versus demonstrating plasma membrane, cytosolic, perinuclear, and nuclear staining in cells derived from P2Y_6_^+/+^ mice. Furthermore, the presence of β-catenin in the nucleus correlated with abnormal expression of proto-oncogene c-MYC. These data indicate that the expression of P2Y_6_ is associated with the localization of β-catenin, which is related to increased proliferative and invasive phenotypes and poor outcomes in CRC patients [193]. However, the authors suggest a cautious interpretation of their results, as the sample size was small, and further investigations are needed.

The P2Y_12_ receptor in platelets is well known for its role in arterial thrombosis, and antagonists are used for the prevention and treatment of cardiovascular diseases [194]. The efficacy of one of P2Y_12_ antagonist (ticagrelor) therapy for breast and CRC was tested, as platelets participate in cancer-associated thrombosis and metastasis [195]. Ticagrelor significantly inhibited formation of tumor cell-induced platelet aggregation in vitro and reduced platelet aggregation and activation in patients. These findings suggest the role of P2Y_12_ in cancer-associated platelet aggregation and put forth P2Y_12_ inhibition as a potential therapy for patients with a high risk of cancer-associated thrombosis [195].

## 5. Concluding Remarks

In summary, the CD39/CD73 axis in tumor-associated immune cells promotes immune exhaustion, impairment of antitumor immune activity, and increased CRC progression. CD73 overexpression in cancer cells is associated with tumor growth, chemoresistance, and decreased patient survival. In contrast, decreased CD73 expression is associated with lower immunosuppressive ADO levels and decreased M2-*like* macrophage polarization. Multiple studies have highlighted roles for the P1R family in CRC. A1R expression was associated with tumor regression, A2AR with increased PD-L1 expression, tumor progression, and lower survival, and A2B with increased proliferation, but also increased apoptosis of tumor cells. A role for P2R family members in CRC progression has also been described. P2X5 and P2X7 expression were associated with decreased survival and increased tumor progression, yet one conflicting study indicated a potential protective role for P2X7 against CRC development. P2Y_2_ and P2Y_6_ promote chemoresistance, and P2Y_12_ increases platelet aggregation in tumor-associated thrombosis. Extracellular ATP exhibits dual effects, either reducing tumor cell proliferation or favoring chemoresistance in a concentration-dependent fashion. Figure 3 summarizes the main findings of the current review.

Taken together, these studies indicate that purinergic signaling is active in both tumor and tumor-associated immune cells. The CD39/CD73 ectonucleotidase axis and purinergic receptors exhibit prognostic value and have the potential to be utilized as therapeutic targets. However, more studies are needed prior to the implementation of purinergic-based therapies in clinical practice.

Limitations of the current review include the sole inclusion of papers written in English and published from 2009 until present. It is possible that critical pieces of older literature or relevant works published in other languages were excluded. A second limitation is the lack of access to one article which met inclusion criteria, which was therefore excluded.

## 6. Open Questions

Numerous studies have investigated the participation of T cells in CRC progression. However, little is known about the role of other immune cell types, in particular innate immune cells, such as macrophages, neutrophils, and MDSCs. While macrophages have been robustly correlated with purinergic signaling within the TME, their role has been neglected in CRC. Furthermore, correlations of immune cell exhaustion and purinergic signaling should be performed in CRC. Considering that the TME is fundamental to tumor progression and is unique in CRC due to a non-sterile environment, the microbiome also has the potential to influence tumor growth and response to chemo/radiotherapy. However, only a few studies examining correlations between microbiota and purinergic signaling in CRC were found. An improved understanding of the influence of ectonucleotidases and purinoceptors on the CRC TME is needed in order to enable potential therapeutic targeting of these processes.

As there are flaws to the TNM system for the staging of CRC patients, purinergic signaling may offer additional biomarkers for an improved understanding of tumor progression and the response to chemo/radio/immunotherapy in CRC.

The majority of studies that analyzed ectonucleotidase expression did not demonstrate cell-specific location within tumor, stromal, or immune cells. It is necessary to clarify this as cell-specific expression profiles likely correlate with better or worse prognosis for patients. Additionally, due to difficulty analyzing the functions of these receptors, many studies have failed to prove that the effects shown were mediated by purinoceptors. Therefore, much work is needed to better illuminate the role of purinoceptors in the creation of an immunosuppressive TME in CRC.

## Figures and Tables

**Figure 1 cancers-14-04887-f001:**
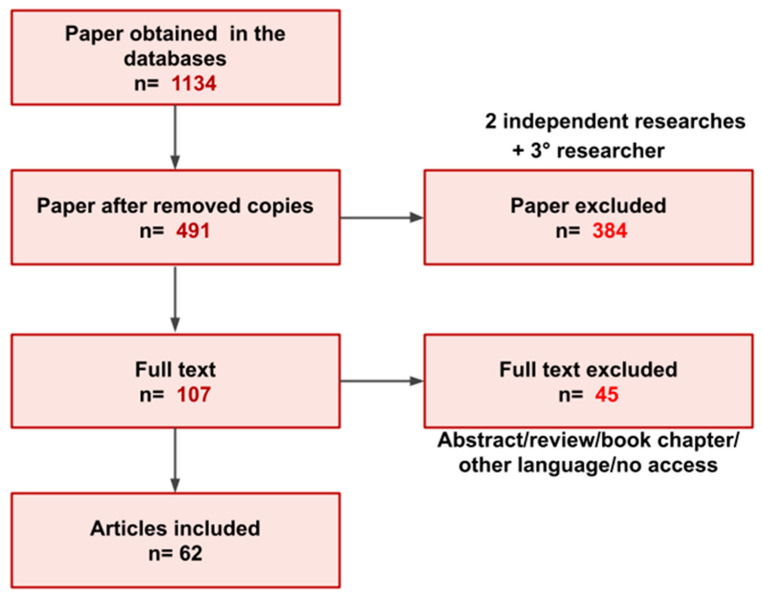
Overview of the paper selection process for inclusion in the present review.

**Figure 2 cancers-14-04887-f002:**
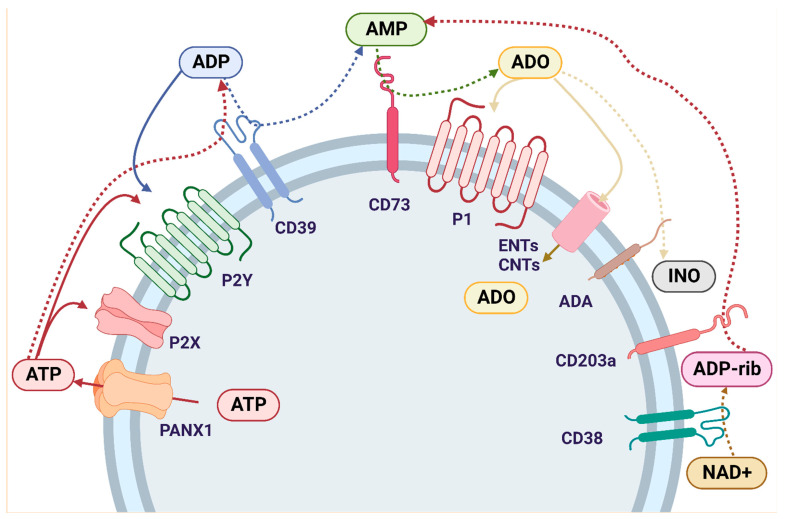
Purinergic signaling cascade: Intracellular ATP is externalized by the PNX1 channel and activates P2 receptors. P2Rs are divided into two major categories: P2X1-7 and P2Y_1,2,4,6,11–14_. ATP is hydrolyzed by E-NTPDase1/CD39 to ADP, which also binds to P2YR and is further hydrolyzed by CD39 to AMP. AMP is also formed through an alternative pathway that involves the enzymes NAD+-glycohydrolase/CD38, which converts NAD+ to ADP-ribose, and NPP1/CD203a, which metabolizes ADP-ribose to AMP. Once AMP is formed by the canonical and/or non-canonical pathway, it is hydrolyzed by ecto-5′-nucleotidase/CD73 to ADO, thus connecting both adenosinergic pathways. ADO binds to the P1 receptors, which are divided into A1, A2A, A2B and A3, each with different ligand affinities. ADO is internalized into the cell through ENTs or CNTs and/or it is deaminated to INO by ADA. Abbreviations: ATP (adenosine triphosphate); ADP (adenosine diphosphate); AMP (adenosine monophosphate); ADO (adenosine); INO (inosine); NAD+ (nicotinamide adenine dinucleotide); PNX1 (Pannexin-1); P2X (ionotropic purinergic receptors); P2Y (metabotropic purinergic receptors); P1 (adenosine metabotropic receptors); E-NTPDase1/CD39 (ecto-nucleoside triphosphate diphosphohydrolase 1); ENTs (nucleoside equilibrative transporters); CNTs (concentrative nucleoside transporters); ADA (adenosine deaminase).

**Figure 3 cancers-14-04887-f003:**
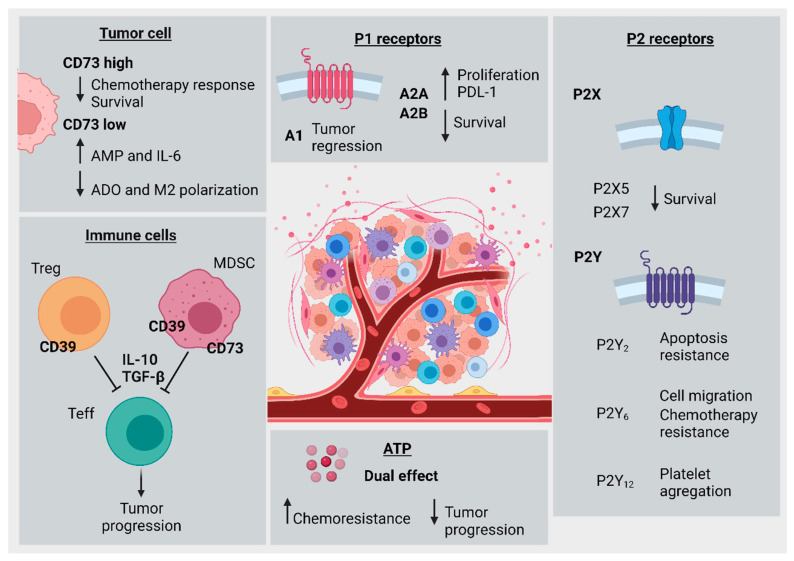
Hallmarks of purinergic signaling in colorectal cancer progression. Tumor cell (**top left**): high levels of CD73 expression were associated with decreased chemotherapy response and patient survival. Low CD73 expression was associated with decreased ADO levels and impairment of M2-macrophage polarization, along with increased AMP and IL-6 levels. Immune cells (**bottom left**): CD39 expression in T regulatory cells or MDSCs was associated with T effector cell inhibition by cytokines like IL-10 and TGF-β, leading to subsequently increased tumor progression. Extracellular ATP (**bottom middle**): ATP levels have a dual effect, and are associated both with increased chemoresistance and, alternatively, reduced CRC cell viability and proliferation. P2 Receptors (**bottom right**): P2X5 and P2X7 were both associated with decreased patient survival. P2Y_2_ was associated with increased resistance to apoptosis, P2Y_6_ was associated with increased cell migration and chemotherapy resistance, and P2Y_12_ was associated with increased platelet aggregation. P1 Receptors (**top right**): A1 receptor was correlated with tumor regression, A2A and A2B were correlated with increased PD-L1 expression and cellular proliferation, and both were corresponded to decreased patient survival.

**Table 1 cancers-14-04887-t001:** Overview of purinergic signaling participation on colorectal cancer biology—preclinical and clinical analysis.

Reference	Type of Study	Murine or Cell Model	Target	Results
Wen et al., 2019	In vitro/In vivo	CT29/BALB/c	ATP	Increase of ATP levels in the microenvironment
Yagushi et al., 2010	In vitro	Caco-2	ATP	ATP-mediated PKC inhibition via P2R sensitization
Vinette et al., 2015	In vitro	Caco-2	ATP	ATP-mediated increase of MRP2 expression via P2R sensitization
Kim et al., 2019	Human tumor (N = 136)	CRC (stage III)	ATP	ATP sensitivity of patients is directly correlated with better response to chemotherapy
Dillard et al., 2021	In vitro	HT29, HCT116, LS513 and LS174T	ATP and ADO	ATP induces cell death in CRC cell lines
Kunzli et al., 2011	In vivo	CD39 transgenic, CD39+/−, and CD39 wild-type mice	CD39	Increased CD39 expression in endothelium, stromal and mononuclear infiltrating tumor cells High P2Y_2_ expression in metastatic liver tumors
	Tumor (N = 63)/adjacent tissue (N = 13)	CRC stage		Lower expression of CD39 and P2Y_2_ in tumor tissue at initial stages of CRC compared to metastatic tumors
McCarthy et al., 2013	In vivo/In situ	HCT116 and HCT15	CD39L4	Lower levels of ATP in transfected mt-PCPH and myc-tagged PCPH cells. However, the NTPDase activity of mt-PCPH was undetectable.
Parodi et al., 2013	Tumor (N = 2 CRC/blood donor (13)	Renal, Bladder, CRC stage (III)	CD39	High CD39 expression in circulating cells from patientsHigh CD39 expression in intratumoral CD8+ Tregs
Zhang et al., 2013	Blood (N = 64)/tumor (N = 5)/in vitro	CRC (I, II, III, IV) cells from blood	CD39	Increased MDSCs correlates with tumor metastasis and increased stage MDSCs CD39^high^ inhibits CD3^+^ T cell proliferation
Scurr et al., 2014	Tumor/ adjacent tissue, blood/blood healthy donor (N = 40)	CRC stage I, II (majority), III	CD39	Increased expression of CD39 in both FOXP3^+^ and FOXP3^−^ tumor-associated Tregs when compared to circulating Tregs from healthy donors or CRC patients
Sundstrom, 2016	Blood (N = 45) and tumor (N = 7)	-	CD39	Higher expression of CD39 in circulating and tumor-associated Tregs when compared to healthy donors Adenosine reduces T cell migration by impairing the capacity of monocytes to activate the endothelium
Limagne et al., 2016	Blood mCRC (N = 25) and blood healthy donor (N = 20)	CRC stage IV	CD39	PD-L1^high^ CD39^high^ CD73^high^ gMDSC levels are associated with poor prognosis FOLFOX plus bevacizumab decreased gMDSC levels
Timperi et al., 2016	Tumor/adjacent tissue, blood (N = 34)		CD39	Presence of CD39^high^ Tregs with production of IL-17 and IL-1β increased in tumor site The ENPD1 SNP rs10748643 contributes to CD39 expression
Hu et al., 2017	in vitro	SW480	CD39	A2AR and A2BR antagonists blocked the activity of γδTregs CD39+ γδTreg inhibited CD3^+^ T cell proliferation
	Tumor/ adjacent tissue (N = 109), blood donors	-		Tumor-associated CD39^high^ γδTreg is correlated to high CTLA-4, PD-1, FOXP3, IL-10, IL-17A, GM-CSF, TGF-β1, and TNFα production. TGF-β1 induced γδTreg to produce more adenosine
Khaja et al., 2017	Tumor/ adjacent tissue (N = 12)	CRC stages (I,II,III and IV)	CD39	CD4+FOXP3^+^ T cells demonstrated high co-expression of PD-1/CTLA-4 and PD-1/CD39; CD39 was overexpressed in tumors
Zhulai et al., 2018	Blood (N = 42)/blood healthy donors (N = 30)/tumor (N = 5) stage III	Initial (I and II) and advanced (III and IV) staged CRC	CD39	Advanced stage CRC demonstrated increased CD4+CD39^+^ lymphocytes in blood and tumor tissue, as well as a negative correlation with CD3+CD4+ T helper cells and CD3^−^CD19^+^ B cells. Association between FOXP3 and CD39 in CD4+CD25^high^ T cells.
Simoni et al., 2018	Tumor/ adjacent tissue (N = 94)	Stage I, II, III and IV	CD39	Lower CD39 expression in CD8+ bystander TILs than tumor-specific CD8+ TILs. CD39+ was correlated with genes associated with T cell proliferation and exhaustion.
Strasser et al., 2019	Tumor/ adjacent tissue (N = 29), gene data (N = 298)	-	CD39	Higher levels of CD39+Helios+ T cells and pro-inflammatory IFNγ -producing T cells in CRC tissue
Gaibar et al., 2021	Tumor mCRC (N = 57, paraffin)	Stage IV	CD39	Variant allele CD39 patients demonstrated better response to bevacizumab plus chemotherapy, but no changes to OS or PSF
Gallerano et al., 2020	Tumor/ adjacent tissue, blood (N = 60)	Stage I, II, III and IV	CD39	CD8+CD39^high^ T lymphocytes were expressed at higher levels within the tumor at initial stage of CRC (I-II), with high PD-1 expression and lower INF-y production. These were correlated with exhausted T cells and suppressed CD4+ T cell proliferation.
Park et al., 2021	in vitro/in vivo	Balb/c subcutaneously injected in flank with CT26 cells. Intraperitoneally injected POM-1 daily for 2 weeks.	CD39	CD39 inhibitor increased CD11b and Ly6C expression in M1 TAMs and F4/80+ macrophages in vitro. CD39 inhibitor resulted in smaller tumors, increased Ly6C and MHC II in F4/80+ macrophages, increased CD8+ T cells in the spleen, increased CD4+ T cells in the blood, and increased Caspase-3 expression, compared with the saline treatment (control group) in vivo.
Rodin et al., 2021	Tumor/ adjacent tissue (N = 28 male, N = 19 female) 10 cm away from tumor	Stage I (4), II (17), III (26), IV (1)	CD39	CRC infiltrating MAITs (mucosal-associated invariant T cells) have a terminally exhausted phenotype (PD-1^high^Tim-3+CD39+). MAIT cells have reduced polyfunctionality with decreased production of antitumor effector molecules, and blocking PD-1 improved activation of tumor-infiltrating MAIT cells in vitro.
Zhao et al., 2020	In vitro	MC38 and HT29	CD39	Expression level of CD39 in colorectal tumor tissues was higher than in normal tissues. CD39 was also highly expressed in both human and murine colorectal cancer cell lines MC38 and HT29. CD39 inhibitor decreased MC38 cell growth at 48 and 72 h. CD39 inhibitor reduced cell proliferation in a dose-dependent manner.
Zhan et al., 2021	Tumor/ adjacent tissue (N = 129)/ In vivo/In vitro	Stage I, II, III and IV. In vitro: CT26. In vivo: CT26-Vec/Pla2g4a cells transplanted into the caecal wall of BALB/c.	CD39	Left-sided CRC had lower frequency of CD39+γδ Tregs than right-sided CRC. Right sided CRC had increase adenosine level, increased IL-17A production, and decreased IFN-γ–production.
Bonnereau et al., 2022	Tumor/adjacent tissue (N = 44)	Right colon (20), left colon (16), rectum (8). Autologous coculture	CD39/CD73	CD4+/CD8+ T cells in tumors demonstrated increased CD39 expression and decreased CD73 expression in early stage tumors. Conversely, advanced stage tumors demonstrated decreased CD39 expression and increased CD73 expression in T cells. CD39 blockade increased T cell capacity of infiltration tumor spheroid destruction in cocultures
Matsuyama et al., 2010	In vitro	SW48 and SW48LM2	CD73	Reduced CD73 expression in highly liver-metastatic cell line
Wu et al., 2012	Tumor (N = 16 fresh, N = 358 paraffin)	CRC: stage I (N = 54), stage II (N = 147), stage III (N = 124), stage IV (N = 30)	CD73	High CD73 expression in fresh or paraffin CRC tissue
Cushman et al., 2014	Tumor (N = 103)	mCRC and respective primary tumor	CD73	Higher levels of CD73 expression were predictive of improved PFS following cetuximab treatment
Zhang et al., 2015	Tumor/adjacent tissue (N = 90)	CRC: stage I (N = 11), stage II (N = 38), stage III (N = 40), stage IV (N = 1)	CD73	Higher expression of CD73 in both tumor and stromal tissue compared to peritumoral tissue.
Wu et al., 2016	In vitro	RKO, SW480, HCT-15, LoVo and KM12	CD73	CD73 expressed in five CRC cell lines; overexpression of CD73 promoted β-catenin/cyclin D1 and EGFR expression.
	In vivo	CRC human with/without CD73 interference		CD73 increased tumor size and weight
Hatch et al., 2016	Blood (N = 152)/tumor (N = 71)	-	CD73	High plasma levels of CD73 were predictive of shorter OS in all patients. However, high CD73 was correlated with PFS benefit in the KRAS-WT group treated with cetuximab
Xie et al., 2017	In vitro	HEK293T cells, SW480, HCT116, LoVo, CaCo2, HT29, RKO, DLD1, HCT8	CD73	miR-30a has a negative effect in regulating expression of CD73 mRNA and protein levels, leading to decreased proliferation and increased apoptosis of cancer cells.
	In vivo	BALB/c nu/nu mice with injection of SW480 in dorsal skin		Decreased in the mean weight of miR-30a-treated group.
	Tumor/adjacent tissue (N = 27)	-		Lower expression levels of miR-30a and higher expression levels of CD73 within CRC than the corresponding adjacent control tissues.
Sun et al., 2017	In vitro	CT26, RAW 264.7	CD73	CD73 knockdown and ADO receptor antagonists correlated with decreased M2 polarization and decreased tumor cell proliferation
	In vivo	WT BALB/c mice		Mice under dietary restriction demonstrated reduced tumor growth without body weight reduction, along with reduced M2 macrophage polarization
Wang et al., 2019	Blood (N = 232)/Healthy blood donors (N = 158)	Stage I/II (N = 110), stage III/IV (N = 122)	CD73	Higher CD73 expression in CRC patients compared with healthy donor. Correlation between CD73 expression and several worse clinicopathological features. Shorter OS patients with higher CD73 expression.
Liu et al., 2020	In vivo	Colitis-associated tumorigenesis	CD73	CD73 inhibitor led to decreased loss of body weight, decreased number of tumors, longer colon, lower histopathological score, and downregulated expression of CRC tumorigenesis-associated genes. ADO agonist (NECA) demonstrated the opposite effects, and increased TNF-α and IL-6 production.
Yu et al., 2020	In vivo	CD73 null/A2Bnull	CD73	CD73 promoted tumor progression, along with suppression of antitumor immunity. ADO released during cell death binds to A2B, leading to increased CD73, and binds to A2A, leading to immune suppression.
	In vitro	EG7.OVA and MC38		
	Tumor (N = 25)	-		
Messaoudi et al., 2020	Microarray CRCm (N = 251), blood (N = 193)	CRC stage IV	CD73	CD73^high^ tumors were associated with more aggressive CRC metastasis to liver, poorer response to preoperative chemotherapy presence to mutation in KRAS, shorter time to recurrence, and reduced disease-specific survival.
Kim et al., 2021	In vitro/In vivo	CT26 implantation in BALB/C; colitis-associated cancer model mouse by injection of azoxymethane and Dextran Sulfate Sodium	CD73	Nt5e and Entpd1 expression affects TCR diversity and transcriptional profiles of T cells; CD73 inhibitor (AB680) improved the anticancer functions of immunosuppressed cells, including Treg and exhausted T cells, and caused increased activation of CD8+ T cells.
Lai et al., 2021	In vitro/In vivo Tumor and blood	Male C57BL/6 and P14 TCR transgenic, CD28^−/−^, P14CD28^−/−^ and CD73^−/−^ mice	CD73	CD28^−/−^ mice increase CD73 expression in CD8+ T cells, without differences in CD39 expression, and with increased adenosine level in culture supernatant. CRC tumor and PBL demonstrated CD73 upregulation in Cd28^−/−^/CD8+ T cells. There was reduced cytolytic activity of CD8+ T cells following treatment with supernatant from CD28^−/−^ cells
Ploeg et al., 2021	In vitro	H292, OvCAR3, DLD1, PC-3M and CHO–K1	CD73	Extracellular vesicles derived from cancer cells lines and patients are enriched in CD73. CD73 inhibition in extracellular vesicles leads to reactivate proliferative and cytotoxic capacity of T cells.
Terp et al. 2021	In vitro/human samples (dataset)	HCT116, SKBr3, CT26.CL25 (CT26), A549, PC9, MC38 and 4T1.2 (4T1)	CD73	CD73 expression was significantly higher in tumors of nonresponders vs. responders to anti-EGFR treatment. Decreased PFS in patients with CD73^high^ vs. CD73^low^ tumors.
Lan et al., 2017	In vitro	HCT116 and SW480	A1	Metformin induced increased A1 expression, suppressed proliferation, and induced apoptosis in both CRC cells in an AMPK-mTOR pathway dependent manner.
Wu et al., 2019	Tumor/adjacent tissue (N = 204)	CRC stage I/II (N = 106) and stage III/IV (N = 98)	A2A	Higher A2A expression in tumor than non-tumor tissue was correlated with tumor size, depth of tumor invasion, and increased TNM stage and PD-L1 expression.
Kitsou et al., 2020	In silico (N = 453)	RNA seq and clinicopathological data	A2A	A2A demonstrated lower expression in CRC compared to normal tissue and was not correlated to OS. In colorectal adenocarcinoma, TIL load was positively correlated to A2A expression.
Ma et al., 2010	In vitro	DLD1, SW480, HCT-15, LOVO, COLO205	A2B	Higher A2B expression than A1, A2A, and A3 in tissue samples and in cell lines was increased in hypoxic conditions. Inhibition of A2B decreased cell growth.
	Tumor (N = 88)/adjacent tissue (N = 62)	-		
Long et al., 2013	In vitro	Saos-2, Phoenix Eco, U2OS, HCT116	A2B	A2B is upregulated directly by p53, which is activated by cellular stress and can induce cell death by apoptosis. This was demonstrated in hypoxic conditions and during response to chemotherapy.
Molck et al., 2016	In vitro	DLD1, SW480, CPP14, HEK293T	A2B	A2B antagonist increased mitochondrial oxygen consumption and intracellular ROS levels
Balber et al., 2017	In vitro	HT-29 and CHO-K1	A3	High A3 expression in HT-29 cells
	In vivo	Immunodeficient CB17-SCID		No difference between the CHO-K1 and HT-29 cells xenografts.
	Tumor/adjacent tissue (N = 2)	-		[^18^F]FE@SUPPY accumulation was higher in CRC than in healthy tissue and corresponded to higher expression of A3
Marucci et al., 2018	In vitro	Caco-2, PC3, HepG2, CHO	A3	A3 agonist inhibited Caco-2 cell growth and migration, promoted apoptosis and increased ROS levels. However, A3 knockdown did not prevent agonist effects.
			**P2X**	
Gao et al., 2018	In silico (N = 206)	-	P2X5	P2X5 expression was correlated with worse prognosis and expression levels were higher in the high-risk group.
Janakiram et al., 2015	In vivo	Rag1^−/−^ Apc^Min/+^	P2X7	Increased expression of P2X7R upon Treg transfer and NK cell depletion led to increased tumor cell proliferation, and increased intestinal tumor formation and growth.
Hofman et al., 2015	In vivo	Disrupted P2X7 gene	P2X7	P2X7blockade stimulated Treg accumulation, reduced colonic inflammation and increased tumor proliferation. This was associated with elevated expression of TGFB1.
Quian et al., 2017	Tumor (N = 12 fresh/N = 116 paraffin)	Stage I (N = 31), stage II (N = 36), stage III (N = 44) and stage IV (N = 5)	P2X7	P2X7 was increased in tumor tissue and correlated higher TNM stage.
Zhang et al., 2019	In vitro	NCM460, HCT116, SW480, SW620	P2X7	Higher expression of P2X7was demonstrated in CRC cell lines, even higher in mCRC cell line, compared to cell lines derived from normal colon cell.
	Tumor/adjacent tissue (N = 97)	Stage I (N = 16), stage II (N = 30), stage III (N = 41) and stage IV (N = 10)		Higher expression of P2X7in 56 patients with CRC versus no change in 41 patients, compared to adjacent normal column tissues. Higher expression of P2X7in tumor tissue was associated with more advanced disease and shorter survival. P2X7expression was higher in mCRC compared to primary colorectal tissues.
Zhang et al., 2021	In vitro/In vivo	HT116, SW620	P2X7	P2X7inhibitor (A438079) inhibits CRC cells line proliferation, invasion, and migration and promotes apoptosis by Bcl-2/caspase9/caspase3 pathway. In vivo, P2X7inhibitor inhibits tumor growth.
Yang et al., 2020	In vitro/In vivo	Injected 10 μL of 2 × 10^6^ CT26-Con or CT26-mP2X7R cells into the subserosa of the caecum	P2X7	P2X7R promoted proliferation, migrated, invasion, and increased the number of tumorspheres of CRC cells in vitro. P2X7overexpression increased the growth and weight of tumors, infiltration of macrophages, TAM recruitment, and stimulation of angiogenesis in vivo.
Bernardazzi et al., 2022	In vivo	P2X7^+/+^ and P2X7^−/−^ mice AOM/DSS-treated	P2X7	P2X7^+/+^ mice demonstrated increased TNF-alpha, IL-17A, and IL-6 following AOM/DSS treatment compared with P2X7^−/−^ mice. The P2X7antagonist (A740003) increased survival and decreased symptoms of tumorigenesis, including body weight loss, shortened colon length, and number of polyps/tumors. Overall, A740003 prevented tumor development in the P2X7^+/+^ group. No tumor formation was observed in the P2X7^−/−^ group. Dissimilarity in the fecal microbiota was observed between the A740003-treated and untreated AOM/DSS-induced P2X7R^+/+^ mice, and between the P2X7R^+/+^ control mice and P2X7^−/−^ control mice.
			**P2Y**	
Limami et al., 2012	In vitro	HT-29	P2Y_2_	Ursolic acid induced an increase in intracellular ATP and in P2Y_2_ mRNA. p38 activation was dependent on P2Y_2_ activation.
Placet et al., 2018	In vitro	HT-29	P2Y_6_	P2Y6R agonist prevents apoptosis. Stimulation of P2Y6R prior to 5-FU treatment provides protection.
	In vivo	P2Y6^−/−^ or P2Y6^+/+^ mice		Decreased number and volume of CRC tumors in P2ry6^−/−^ mice P2Y6R increased expression of XIAP and correlated with AKT phosphorylation and resistance to 5-FU
Girard et al., 2020	In vitro	Caco-2	P2Y_6_	P2Y_6_ increased cell migration, through PKCα that stabilizes the actin cytoskeleton
Wright et al., 2020	In vitro	HT-29	P2Y_12_	Reduced cell aggregation and adhesion
	Oncological patients blood (N = 6)/Healthy donors blood (N = 22)	mCRC		Higher levels of spontaneous platelet aggregation and P-selectin expression in mCRC tissues

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
