# Peer review of "Colorectal Cancer and Purinergic Signalling: An Overview"

_cancers, 2022, doi:10.3390/cancers14194887_

Round 1

Reviewer 1 Report

The review on the role of purinergic signaling in the field of CRC (initiation/progression) is presented from a broad point of view, including recent reports that study the role of this signaling pathway in different cell types. Special interest is placed on immune cells and the expression of receptors involved in purinergic sensing.

The graphic representations are suitable to facilitate the understanding of the topics covered in the text. In addition, the authors provide closing comments and open-ended question sections that are highly relevant to establishing the state of the art in the field.

Minor points:

Authors must proofread the text. At several points, the sentences lack cohesion (for example, on page 4, point 2.1)

References should be checked (reference 25 is missing).

Author Response

Reviewer 1

Comments and Suggestions for Authors

1. The review on the role of purinergic signaling in the field of CRC (initiation/progression) is presented from a broad point of view, including recent reports that study the role of this signaling pathway in different cell types. Special interest is placed on immune cells and the expression of receptors involved in purinergic sensing.

The graphic representations are suitable to facilitate the understanding of the topics covered in the text. In addition, the authors provide closing comments and open-ended question sections that are highly relevant to establishing the state of the art in the field.

2. Minor points: Authors must proofread the text. At several points, the sentences lack cohesion (for example, on page 4, point 2.1)

The manuscript was carefully reviewed in order to improve English written style and quality. Thank you for your consideration.

 3. References should be checked (reference 25 is missing).

The reference section was revised and corrected. Thank you.

Reviewer 2 Report

The comprehension of the mechanisms regulating the communication between immune and cancer cells within the tumor microenvironment is a relevant topic in oncology given the impact of immune dysfunctions on cancer progression.

The review article by Roliano and colleagues focuses on the contribution of the purinergic signaling to this cross-talk and summarizes the main achievements on the role of the different components of this pathway in colorectal cancer. Some recent review articles on single members of the pathway, however, are already available.

Although interesting, some points are not clear and the manuscript would benefit from some revisions. Specifically:

- The English language and style throughout the manuscript need a careful revision 

- It is not clear why papers published after November 2021 (almost a year ago) were not included. In this regard, the paper by Bernardazzi C et al (Int J Mol Sci 2022), on the cooperation between P2X7R signaling and microbiota, as well as any others, should be discussed and quoted in the manuscript

- Section 3 could be shortened and I would suggest to reorganize it into “innate and adaptive immune cells”. A brief mention of the involvement of other immune cell subsets (dendritic cells, NK cells, …) should also be made

- The text is often very long and difficult to follow, individual paragraphs should be slimmed down by reducing some details

Abbreviations should appear the first time they are quoted 

- Figures 1 and 2 are not quoted in the text

Author Response

Reviewer 2

The comprehension of the mechanisms regulating the communication between immune and cancer cells within the tumor microenvironment is a relevant topic in oncology given the impact of immune dysfunctions on cancer progression.

The review article by Roliano and colleagues focuses on the contribution of the purinergic signaling to this cross-talk and summarizes the main achievements on the role of the different components of this pathway in colorectal cancer. Some recent review articles on single members of the pathway, however, are already available. Although interesting, some points are not clear and the manuscript would benefit from some revisions. Specifically:

1. The English language and style throughout the manuscript need a careful revision

The revision of English language and style was performed by a native English speaker. Thank you for your suggestion.

2. It is not clear why papers published after November 2021 (almost a year ago) were not included. In this regard, the paper by Bernardazzi C et al (Int J Mol Sci 2022), on the cooperation between P2X7R signaling and microbiota, as well as any others, should be discussed and quoted in the manuscript.

Relevant articles, such as Bernardazzi et al 2022, published in 2022 were included in the manuscript. Thank you.

3. Section 3 could be shortened and I would suggest to reorganize it into “innate and adaptive immune cells”. A brief mention of the involvement of other immune cell subsets (dendritic cells, NK cells, …) should also be made.

As suggested by Reviewer 2, the section 3 was shortened and the topics were presented as follow: Influence of the gut microbiome on tumorigenesis; Macrophages and neutrophils; Myeloid-derived suppressor cells; Lymphocytes. We hope that the Reviewer 2 appreciates this topic presentation.

4. The text is often very long and difficult to follow, individual paragraphs should be slimmed down by reducing some details

Thank you for the suggestion. We completely agree with the comment. The text of the submitted Review was extensively revised in order to clarify the concepts and to better explore the topics. We hope that the Reviewer 2 appreciates the revised version of the manuscript.

5. Abbreviations should appear the first time they are quoted.

The correction was done. Thank you.

6. Figures 1 and 2 are not quoted in the text

The correction was done. Thank you.

Round 2

Reviewer 2 Report

The manuscript organization and English style have been improved. No further changes are required.